# Drug prescribing and dispensing practices in regional and national referral hospitals of Eritrea: Evaluation with WHO/INRUD core drug use indicators

Senai Mihreteab Siele[1], Nuru Abdu[1]*, Mismay Ghebrehiwet[2], M. Raouf Hamed[3], Eyasu H. Tesfamariam[4]

1 School of Pharmacy, Asmara College of Health Sciences, Asmara, Eritrea, 2 Advisor to the Minister of Health, Ministry of Health, Asmara, Eritrea, 3 National Organization for Drug Control and Research, Cairo, Egypt, 4 Department of Statistics, Biostatistics and Epidemiology Unit, Department of Statistics, Mai Nefhi College of Science, Mai Nefhi, Eritrea

* pharmacistnuru@gmail.com

**Data Availability Statement:** All relevant data are within the manuscript and its Supporting Information files.

## Abstract

Rational use of medicine (RUM) for all medical conditions is crucial in attaining quality of healthcare and medical care for patients and the community as a whole. However, the actual medicine use pattern is not consistent with that of the World Health Organization (WHO) guideline and is often irrational in many healthcare setting, particularly in developing countries. Thus, the aim of the study was to evaluate rational medicine use based on WHO/International Network of Rational Use of Drugs (INRUD) core drug use indicators in Eritrean National and Regional Referral hospitals. A descriptive and cross-sectional approach was used to conduct the study. A sample of 4800 (600 from each hospital) outpatient prescriptions from all disciplines were systematically reviewed to assess the prescribing indicators. A total of 1600 (200 from each hospital) randomly selected patients were observed for patient indicators and all pharmacy personnel were interviewed to obtain the required information for facility-specific indicators. Data were collected using retrospective and prospective structured observational checklist between September and January, 2018. Descriptive statistics, Welch's robust test of means and Duncan's post hoc test were performed using IBM SPSS (version 22). The average number of medicines per prescription was 1.78 (SD = 0.79). Prescriptions that contained antibiotic and injectable were 54.50% and 6.60%, respectively. Besides, the percentage of medicines prescribed by generic name and from an essential medicine list (EML) was 98.86% and 94.73%, respectively. The overall average consultation and dispensing time were 5.46 minutes (SD = 3.86) and 36.49 seconds (SD = 46.83), respectively. Moreover, 87.32% of the prescribed medicines were actually dispensed. Only 68.24% of prescriptions were adequately labelled and 78.85% patients knew about the dosage of the medicine(s) in their prescriptions. More than half (66.7%) of the key medicines were available in stock. All the hospitals used the national medicine list but none of them had their own medicine list or guideline. In conclusion, majority of WHO stated core drug use indicators were not fulfilled by the eight hospitals. The results of this study suggest

**Funding:** The study was funded by the National Higher Education and Research Institute of Eritrea. The funders had no role in study design, data collection and analysis, decision to publish, or preparation of the manuscript.

**Competing interests:** The authors have declared that no competing interests exist.

that a mix of policies needs to be implemented to make medicines more accessible and used in a more rational way.

## Introduction

Rational use of medicine (RUM) is an essential element in achieving quality of health care for patients and the community. The WHO defines rational use of medicine as providing the right medicine, for the right patient at the right dose, for the right duration and at the lowest possible cost to them and their community [1]. WHO estimates that more than half of all medicines are prescribed, dispensed or sold inappropriately, and that half of all patients fail to take them properly. The overuse, underuse or misuse of medicines results in wastage of scarce resources and widespread health hazards [2].

Irrational use of medicines is a global burden. A number of factors that lead to an irrational use of medicines are polypharmacy, inadequate dosage of antibiotics, use of antibiotics for viral infections, over-use of injections when oral medication can be more suitable [3, 4]. Most physicians would vouch for having observed this in their day-to-day practice but there is no dearth of hard evidence to reinforce this impression.

WHO in collaboration with the International Network of Rational Use of Drugs (INRUD) has developed a manual that defines a limited number of objective measures that can describe the medicine situation in a country, region or individual health facilities. Such measures or indicators will allow health planners, managers and researchers to make basic comparisons between situations in different areas or at different times. Drug use indicators can be used to identify general prescribing and quality of care problems at primary health care facilities thereby enhancing the rational use of medicines.

Eritrea, a developing country in the Horn of Africa, encountered with the challenges of irrational use of medicines in parallel with the rest of the world. Despite the commitment towards ensuring RUM is highlighted in the national drug policy of the Ministry of Health, numerous studies reported that irrational use of medicines still exists in the country [5–7]. This study assessed the rational use of medicines in all regional and national referral hospitals of the country.

## Materials and methods

### Study design and setting

A facility-based cross-sectional study with a quantitative approach was conducted in all the regional (zonal) and two national referral hospitals of Eritrea. The study was conducted from September 1, 2017 to January 31, 2018. Retrospective cross-sectional study was used to evaluate prescribing indicators while prospective cross-sectional study design was employed for patient care and facility indicators.

The provision of health services in Eritrea has been provided through a three tier or level system which include primary, secondary and tertiary level of services. The primary level services includes community-based health services, health station and health center. The secondary level includes a regional (zonal) referral hospital and a second contact hospital within a region (zoba). Moreover, a tertiary level comprises of a national referral hospital. Eritrea has a total of six regions (zobas) namely: Anseba, Debub, Debubawi Keih Bahri, Gash-Barka, Maekel, and Semenawi Keih Bahri. Each region (zoba) has its own referral hospital. Moreover, Eritrea has also four national referral hospitals. This study was conducted in two national referral

hospitals namely: Orotta and Halibet and six regional referral hospitals: Barentu (Gash-Barka region), Mendefera (Debub region), Ghindae (Semenawi Keih Bahri region), Assab (Debubawi Keih Bahri region), Keren (Anseba region) and Hazhaz (Maekel region).

## Study population

All outpatient prescriptions dispensed from January 1, 2017 to December 31, 2017 (prescribing indicators); patient attendants and their prescriptions in the outpatient departments (OPDs) of the selected hospitals from September 1 to November 30, 2018 (patient care indicators) and medicines under Eritrean essential medicine list (EML) of 2015 were included. Nevertheless, prescriptions that contain any item apart from a pharmaceutical and patient attendants outside the normal employment hours were not included in the study.

## Sampling

As per the WHO recommendation, 600 prescribing encounters were taken from each hospital to assess the prescribing practices. As a result, a total of 4,800 prescriptions were investigated in the study. To minimize the sampling bias (seasonal alterations or supply cycle of medicines), the encounters per year were uniformly divided into four quarters and 150 prescriptions were randomly selected from each quarter, irrespective of acute or chronic illnesses, including a mixture of health conditions and a range of patient ages. Then, systematic random sampling was used once sampling frame had been developed by arranging the study population in chronological order of prescription.

Moreover, based on WHO criteria, at least 100 outpatient attendants (encounters) were recommended in individual health facility [8]. Therefore, to get a more reliable outcome 200 patients were assessed in each hospital after spreading them throughout the clinic hours [9]. The patient attendants with their prescriptions in OPDs were sampled by convenient sampling technique prospectively.

As for the assessment of the health facility indicators, key medicines were selected from each hospital as per WHO recommendation which is a minimum of 15 essential medicines in each health facility [8]. These key medicines being used for the management of the leading diseases of the respective hospitals were selected by communicating with prescribers and dispensers and reviewing national guideline [9]. All available pharmacy personnel were invited to participate in the study and the consented participants were interviewed to obtain the required information [8].

## Data collection tools and approach

Data were collected using structured checklists for prescribing, patient care and health facility indicators. Data regarding prescribing indicators were taken from sampled prescription records retrospectively and filled in structured checklist accordingly by careful observation.

Patient prescriptions were used as a reference to check the patient knowledge on how to take the correct dosage of a medicine. A stop watch was used to determine the health care providers-patient interaction time (consultation and dispensing time). Data about patient care indicators were taken from patient attendants and their prescriptions in OPD during the period of data collection prospectively and were recorded in an observational checklist. Among patient care indicators, data of patient knowledge on how they take a correct dosage were collected through face to face interview and recorded as 1 or 0 for each patient. In addition, the availability of key/essential medicines, were evaluated in OPD, and was recorded in the facility indicator form.

## Variable measurement

To evaluate the rational medicine use comprehensively, Index of Rational Drug Prescribing (IRDP), Index of Rational Patient- Care Drug Use (IRPCDU), and Index of Rational Facility-Specific Drug Use (IRFSDU) were developed by Zhang and Zhi for a comprehensive appraisal of medical care [4, 10]. For the calculation of non-polypharmacy, rational antibiotic use and injection safety indices, the following formula was used;

$$\text{Index} = \frac{\text{Optimal Value (WHO standard)}}{\text{Observed value}}$$

The optimal values for calculating the indices of non-polypharmacy, rational antibiotic use and injection safety were taken as 1.8, 26.8 and 24.1, respectively.

All other indices (index of generic prescribing, index of prescribing from an essential medicine list (EML), consultation time index, dispensing time index, index of medicines actually dispensed, index of labelling of medicines, index of patients' knowledge, index of EML availability and index of key medicines availability in stock) was calculated by the following formula;

$$\text{Index} = \frac{\text{Observed Value}}{\text{Optimal value (WHO standard)}}$$

The optimal values for the calculation of indices for generic prescribing, medicine prescribed from EML, medicines actually dispensed, patient knowledge of correct doses, labelling of medicines and availability of key medicines were taken as 100. Besides, the optimal value for the calculation of indices for consultation and dispensing time were taken as 10 minutes and 90 seconds, respectively.

The Index of Rational Drug Prescribing (IRDP) was calculated for all hospitals by adding the index values of all the prescribing indicators. Similarly, the Index of Rational Patient-Care Drug Use (IRPCDU) and the Index of Rational Facility- Specific Drug Use (IRFSDU) were calculated. Based on the IRDP, IRPCDU and IRFSDU values, the hospitals were ranked from 1 to 8 within each category (rank 1 for the higher value and rank 8 for the lower value). The optimal index for all the indicators is one. As the value of the optimal index is closer to one, the more rational the medicine use indicator.

Finally, the Index of Rational Drug Use (IRDU) was calculated for all hospitals by adding up the total of IRDP, IRPCDU and IRFSDU. Moreover, a rank was assigned to each hospital based on the IRDU value.

The country performance indicator for drug prescribing, patient-care and facility-specific was calculated using the same approach with the above-mentioned formulas. For instance, the country performance indicator of generic prescribing was measured by dividing the observed value by the optimal value (100%). The observed value was taken as the average value of the generic prescribing across the eight hospitals. Table 1 displays the WHO optimal values of the core drug use indicators

## Data quality

To ensure data consistency in results all data collectors were trained together and then allowed to practice together in Orotta National Referral Hospital and Hazhaz Regional Referral Hospital. This step provides an opportunity to identify and solve unforeseen problems. It also allowed to make realistic estimate of the time required for collecting data at each site.

**Table 1. Core drug use indicators and their optimal values.**

| Core drug use indicators | WHO Optimal values [8, 11] | Optimal Index [10] |
|---|---|---|
| **Prescribing Indicators** | | |
| Average number of medicines prescribed per patient encounter | 1.6–1.8 | 1 |
| Percentage of medicines prescribed by generic name | 100 | 1 |
| Percentage of encounters with an antibiotic prescribed | 20.0–26.8 | 1 |
| Percentage of encounters with an injection prescribed | 13.4–24.1 | 1 |
| Percentage of medicines prescribed from essential medicines list or formulary | 100 | 1 |
| **Patient-Care Indicators** | | |
| Average consultation time (minutes) | ≥10 | 1 |
| Average dispensing time (seconds) | ≥90 | 1 |
| Percentage of medicines actually dispensed | 100 | 1 |
| Percentage of patients with knowledge of correct doses | 100 | 1 |
| Percentage of drugs adequately labelled | 100 | 1 |
| **Facility-Specific Indicators** | | |
| Availability of essential medicines list or formulary to practitioners | 100 | 1 |
| Availability of key essential medicines | 100 | 1 |

## Statistical analysis

Data entry and analyses were conducted using SPSS (Version 22.0). Mean (SD) or median (IQR) was used to make descriptive analysis for the continuous variables, while frequency (percent) was used for the qualitative ones. Welch's robust test of means was used to assess the possible differences in mean consultation time and mean dispensing time across the health facilities. Subsequently, grouping of the health facilities that have similar mean consultation time and mean dispensing time was also performed using Duncan's post-hoc analysis. IRDP, IRPCDU, IRFSDU, and IRDU indices were also computed appropriately and then ranks assigned to make comparisons among the health facilities. The statistical significant was determined by a $p$-value less than 0.05.

## Ethical consideration

A formal written ethical approval form was obtained from Research Ethical Clearance Committee of Asmara College of Health Sciences with a reference number of 019/07/18 and Ministry of Health research ethics and protocol review committee with a reference number of 22/08/18.

Participant information sheet and a written informed consent form were filled by trained data collectors after explaining the purpose of the study to the individual respondents. Confidentiality was assured for all information collected.

## Results

### Prescribing indicators

A total of 8555 drugs were prescribed from the 4800 prescriptions assessed. The average number of drugs per encounter was 1.78 (SD = 0.79). The mean drugs prescribed ranged from 1.62 (SD = 0.70) in Hazhaz regional referral hospital to 2.07 (SD = 0.80) in Keren regional referral hospital. Most of the drugs were prescribed by their generic names (94.86%). The range of percentage of generic prescribing varied from a minimum of 84.01% in Ghindae regional referral

Table 2. Prescribing indicators vis-à-vis WHO core standards.

| Prescribing indicators | Name of Health facility | | | | | | | | | |
|---|---|---|---|---|---|---|---|---|---|---|
| | ONRH M, SD (Md, IQR) | HNRH M, SD (Md, IQR) | HRRH M, SD (Md, IQR) | GRRH M, SD (Md, IQR) | ARRH M, SD (Md, IQR) | KRRH M, SD (Md, IQR) | MRRH M, SD (Md, IQR) | BRRH M, SD (Md, IQR) | Overall M, SD (Md, IQR) | WHO standard[8, 11] |
| Average number of drugs per encounter | 1.92, 0.98 (2,1) | 1.74, 0.70 (2,1) | 1.62, 0.70 (2,1) | 1.65, 0.67 (2,1) | 1.82, 0.87 (2,1) | 2.07, 0.80 (2,1) | 1.72, 0.69 (2,1) | 1.73, 0.75 (2,1) | **1.78, 0.79 (2,1)** | 1.6–1.8 |
| Percentage of encounter with antibiotic | 53 | 44.3 | 63.8 | 53.5 | 57 | 58.3 | 49.8 | 56.3 | **54.5** | 20–26.8 |
| Percentage of encounter with injection | 6.8 | 7.2 | 8.2 | 7.7 | 7.8 | 3.8 | 6.8 | 4.7 | **6.6** | 13.4–24.1 |
| Percentage of medicines prescribed by generic | 97.04 | 94.72 | 95.88 | 84.01 | 95.41 | 95.34 | 97.58 | 98.08 | **94.86** | 100 |
| Percentage of medicines from essential medicine list | 97.04 | 90.01 | 95.37 | 83.4 | 95.05 | 99.76 | 97.38 | 98.08 | **94.73** | 100 |

**Note:** ONRH = Orotta National Referral Hospital, HNRH = Halibet National Referral Hospital, HRRH = Hazhaz Regional Referral Hospital, GRRH = Ghindae Regional Referral Hospital, ARRH = Assab Regional Referral Hospital, KRRH = Keren Regional Referral Hospital, MRRH = Mendefera Regional Referral Hospital, BRRH = Barentu Regional Referral Hospital, M = Mean, SD = Standard Deviation, Md = Median, IQR = Interquartile range.

hospital, to as high as 98.08 in Barentu regional referral hospital. Generally, the percentage of drugs prescribed using generic name was above 94.72% except in one hospital. Overall, one or more antibiotics was prescribed in 54.5% of the encounters (n = 2617/4800). Hazhaz regional referral hospital showed the highest percentage of prescriptions with antibiotics (63.8%) followed by Keren (58.3%), Assab (57.0%), and Barentu (56.3%). There were 318 (6.6%) prescriptions that contained at least one injectable medications. Majority of the drugs (94.73%) were prescribed from the Eritrean National List of Medicines (ENLM) [Table 2].

### Patient care indicators

Median consultation time was 4 (Q1 = 2 and Q3 = 8) minutes. Welch's robust test of means showed that there was significant difference in consultation time among the health facilities (F = 89.79, $p<0.0001$). Duncan's grouping has discovered Ghindae as group 1, Hazhaz and Mendefera as group 2, Halibet and Orotta as group 3, Keren as group 4, Assab as group 5, and Barentu as group 6 [Table 3].

The median dispensing time of drugs in all the health facilities was 25 (Q1 = 14, Q3 = 45) seconds. Welch's robust test of means showed that there was significant difference in dispensing time among the health facilities (F = 27.76, p<0.0001). Duncan's grouping has discovered Barentu, Keren, Mendefera and Ghindae as group 1, Keren, Mendefera, Ghindae, Halibet, Hahzaz and Orotta as group 2, Assab as group 3 [Table 4].

Majority of the drugs prescribed (87.32%, n = 2623/3004) from the eight hospitals were actually dispensed. An adequate label includes at least the name and strength of the medicine and written instructions on how to take it. The overall median percent medicines adequately labelled was found to be 68.24%. Moreover, the average percent of patients who know how to take medicines was found to be 78.88% [Table 5].

**Table 3. Grouping of hospitals as per their homogeneity with regards to the consultation time (minutes).**

| Health Facility | Group | | | | | | F (df1, df2) | p-value |
|---|---|---|---|---|---|---|---|---|
| | 1 | 2 | 3 | 4 | 5 | 6 | | |
| GRRH | 2.67 min | | | | | | 89.79 (7, 679.49) | <0.0001 |
| HRRH | | 3.47 min | | | | | | |
| MRRH | | 3.9 min | | | | | | |
| HNRH | | | 5.29 min | | | | | |
| ONRH | | | 5.7min | | | | | |
| KRRH | | | | 6.45 min | | | | |
| ARRH | | | | | 7.31 min | | | |
| BRRH | | | | | | 8.9 min | | |
| p-value | 1 | 0.2 | 0.22 | 1 | 1 | 1 | | |

Note: ONRH = Orotta National Referral Hospital, HNRH = Halibet National Referral Hospital, HRRH = Hazhaz Regional Referral Hospital, GRRH = Ghindae Regional Referral Hospital, ARRH = Assab Regional Referral Hospital, KRRH = Keren Regional Referral Hospital, MRRH = Mendefera Regional Referral Hospital, BRRH = Barentu Regional Referral Hospital, F = Fisher's exact test, df = degree of freedom.

## Health facility indicators

Eritrean National List of Medicines (ENLM) and National formulary was available in all the hospitals. Furthermore, 80.1% of the key essential medicines were in stock during the study period [Table 6].

## Indices of performance indicators

Relatively better IRDP value was observed in Mendefera regional referral hospital (IRDP = 4.488) in comparison to the other health facilities. Similarly, the IRPCDU values indicated that Assab regional referral hospital (IRPCDU = 3.926) showed better results compared with the other health facilities. Hazhaz regional referral hospital (IRFSDU = 1.815) showed better index with regard to facility-specific drug use indicators. Overall, Assab regional referral hospital (IRDU = 10.086) was the best performing hospital in terms of core drug indicators [Table 7].

**Table 4. Grouping of hospitals as per their homogeneity with regards to the dispensing time (seconds).**

| Health Facility | Group | | | F (df1, df2) | p-value |
|---|---|---|---|---|---|
| | 1 | 2 | 3 | | |
| BRRH | 20.77s | | | 27.76 (7, 672.54) | <0.0001 |
| KRRH | 25.55s | 25.55s | | | |
| MRRH | 25.69s | 25.69s | | | |
| GRRH | 27.99s | 27.99s | | | |
| HNRH | | 31.7s | | | |
| HRRH | | 31.76s | | | |
| ONRH | | 32.57s | | | |
| ARRH | | | 95.94s | | |
| p-value | 0.11 | 0.14 | 1.00 | | |

Note: ONRH = Orotta National Referral Hospital, HNRH = Halibet National Referral Hospital, HRRH = Hazhaz Regional Referral Hospital, GRRH = Ghindae Regional Referral Hospital, ARRH = Assab Regional Referral Hospital, KRRH = Keren Regional Referral Hospital, MRRH = Mendefera Regional Referral Hospital, BRRH = Barentu Regional Referral Hospital, F = Fisher's exact test, df = degree of freedom, s = seconds (dispensing time).

**Table 5. Patient care indicators vis-à-vis WHO core standards.**

| Prescribing indicators | Name of Health facility | | | | | | | | | |
|---|---|---|---|---|---|---|---|---|---|---|
| | ONRH M, SD (Md, IQR) | HNRH M, SD (Md, IQR) | HRRH M, SD (Md, IQR) | GRRH M, SD (Md, IQR) | ARRH M, SD (Md, IQR) | KRRH M, SD (Md, IQR) | MRRH M, SD (Md, IQR) | BRRH M, SD (Md, IQR) | Overall M, SD (Md, IQR) | WHO standard [8, 11] |
| Average Consultation Time (minutes) | 5.70, 3.72 (5,5) | 5.29, 3.24 (4,4) | 3.47, 2.68 (3,2) | 2.67, 2.17 (2,1) | 7.31, 3.79 (6,4) | 6.45, 3.65 (6,4.75) | 3.90, 3.25 (3,3) | 8.90. 3.85 (8.5) | **5.46, 3.86 (4,6)** | >10 |
| Average Dispensing Time (seconds) | 32.57. 25.12 (27.50,30.75) | 31.70, 53.91 (15,25) | 31.76, 44.27 (25,25) | 27.99, 16.30 (20,20.75) | 95.94, 82.59 (72,71.75) | 25.55, 21.96 (18.50,18) | 25.69, 14.04 (22,19) | 20.77, 13.59 (16,21) | **36.49, 46.83 (25,31)** | ≥90 |
| Percentage of drugs actually dispensed | 92.56 | 91.03 | 87.95 | 81.21 | 90.99 | 94.09 | 97.00 | 63.16 | **87.32** | 100 |
| Percentage of drugs adequately labelled | 71.05 | 21.16 | 79.01 | 91.81 | 36.96 | 71.99 | 87.64 | 93.33 | **68.24** | 100 |
| Percentage of patient knowledge about dosage of dispensed drugs | 79.5 | 67 | 85.5 | 79 | 91.5 | 84 | 85 | 55.5 | **78.88** | 100 |

**Note:** ONRH = Orotta National Referral Hospital, HNRH = Halibet National Referral Hospital, HRRH = Hazhaz Regional Referral Hospital, GRRH = Ghindae Regional Referral Hospital, ARRH = Assab Regional Referral Hospital, KRRH = Keren Regional Referral Hospital, MRRH = Mendefera Regional Referral Hospital, BRRH = Barentu Regional Referral Hospital, M = Mean, SD = Standard Deviation, Md = Median, IQR = Interquartile range.

Eritrea as a country scored 1 in the indices of non polypharmacy and rational injection safety. Nearly similar scores in generic name index (0.949) and essential drugs list (0.947) were scored in all the health facilities. However, lesser index in rational antibiotic index (0.492) was scored. Combined Index of Rational Drug Prescribing (IRDP) score was 4.388. Score of Index of Rational Patient-Care Drug Use (IRPCDU) was 3.296 which is low compared to the optimal value of 5 [Table 8].

## Discussion

### Prescribing indicators

The average number of medicines per encounter was found to be 1.78. This value in general falls within the frame of WHO standard (1.6–1.8) in outpatient settings [8]. Even though the

**Table 6. Percentage distribution on availability of key essential drugs.**

| Health Facility | Percent |
|---|---|
| Orotta | 80 |
| Halibet | 79 |
| Hazhaz | 81.5 |
| Ghindae | 81.2 |
| Assab | 79.7 |
| Keren | 80 |
| Mendefera | 79.9 |
| Barentu | 79.8 |
| **Overall** | 80.1 |

**Note:** ONRH = Orotta National Referral Hospital, HNRH = Halibet National Referral Hospital, HRRH = Hazhaz Regional Referral Hospital, GRRH = Ghindae Regional Referral Hospital, ARRH = Assab Regional Referral Hospital, KRRH = Keren Regional Referral Hospital, MRRH = Mendefera Regional Referral Hospital, BRRH = Barentu Regional Referral Hospital.

overall average number of medicines per encounter falls within the WHO standard, 15.2% of the prescriptions enclosed three or more drugs. In a previous study conducted in community-chain pharmacies in Asmara, Eritrea; the average number of drugs per encounter was 1.76 which is exactly the same with the current finding [5]. The figure was consistent with studies conducted in different parts of Ethiopia [12–14]. However, it was lower than 4.8 reported from Sri Lanka [15], 2.9 reported from Kenya [16], 2.9 from Kuwait [17], 2.4 from Saudi Arabia [18] and 2.04 from China [19]. This difference in the number of drugs prescribed per patient could be due to the variation in the study sites and prescribing habits among various medical disciplines. Even though, polypharmacy was not a problem in our country; prescribers should prescribe the lowest possible number of drugs to treat diseases while avoiding symptomatic treatments as polypharmacy was seen in some prescriptions assessed.

The percentage of encounters with antibiotic(s) prescribed was found to be 54.5% which is twice that of the WHO optimal (20–26.8%) [8]. The highest percentage of encounters with antibiotic was recorded in Hazhaz regional referral hospital (63.8%) and none of the hospitals was within the range of WHOs optimal value [8]. This was similar with previous study conducted in Asmara city (Eritrea), which was 53% [5]. However, it was much higher than studies conducted in UAE 9.8% [20], Sri Lanka [15] and Nepal 28.3% [21]. It was lower than studies conducted in Sudan 63% [22], Uganda 56% [23], Ethiopia 58.1% [12] and Kenya 84.8% [16]. Irrational and overuse of antibiotics may lead to adverse drug reactions, antimicrobial resistance and unnecessary hospital admissions [24, 25]. Such high figure could be due to a number of reasons such as inappropriate prescribing of antibiotics, high prevalence of infectious disease in developing countries resulted in increased number of antibiotics prescription, patient pressure on prescribers and allowing lower health cadres to prescribe medicines. In Eritrea, one physician serves 18,041 patients, to deal with this shortage lower health cadres are allowed to prescribe medicines [5, 25]. This mandates policy makers and program managers to draft and implement strategies that decrease the irrational use of antibiotics.

The percentage of injection prescribed hospitals of Eritrea in the present study was 6.6%, lower than the WHO optimal value (≤10%). This figure was lower than the previous study in Eritrea (7.8%) [5], but higher than a study conducted in India 5.7% [26]. On the other hand, it was much lower than the studies in China (22.9%) [19], Bangladesh (38.1%) [27], Kenya (24.9%) [16] and Sri Lanka (30.1%) [15]. This lower value could be due to high preference of oral route by prescribers as injectable preparations are associated with higher risks of disease transmission, incompliance by patients and are expensive [8].

The WHO clearly mentions prescribing drugs by its generic name as generic prescribing is a safety precaution for patients to adhere to their medications and it eases the communication between healthcare providers and patients. Moreover, generic drugs are less expensive than brand drugs. The average percentage of generic prescribing in the present study was found to be 94.6% with the lowest value seen in Ghindae regional referral hospitals (84.01%). This value was higher than the previous study conducted in Eritrea (83.14%) [5]. Caution should be exercised during result comparison as the previous Eritrean study was conducted in community-chain pharmacies of Asmara city. Moreover, lower results were reported from studies conducted in China (69.2%) [19], KSA (61.2%) [18], Sudan (43.2%) [22], South India (42.9%) [28] and Kenya (27.7%) [16]. However, higher results to the present study were reported in researches done in countries like South Ethiopia (98.7%) [12] and Mozambique (99%). Brand prescribing is associated with unnecessary treatment costs, difficulty of remembering the medication name, accessibility and bioequivalence problems [8]. Therefore, more effort is to be devoted to effectively adhere to generic prescribing in order to promote safe, cost effective and accessible generic drugs.

**Table 7. Index of rational medicine use in Eritrean zonal and national referral hospitals.**

| Performance Indicators | Health Facilities | | | | | | | |
|---|---|---|---|---|---|---|---|---|
| | ONRH | HNRH | HRRH | GRRH | ARRH | KRRH | MRRH | BRRH |
| **Drug Prescribing Indicators** | | | | | | | | |
| Non polypharmacy index | 1.000 | 1.000 | 1.000 | 1.000 | 0.989 | 0.870 | 1.000 | 1.000 |
| Generic name index | 0.970 | 0.947 | 0.959 | 0.840 | 0.954 | 0.953 | 0.976 | 0.981 |
| Rational Antibiotic index | 0.506 | 0.605 | 0.420 | 0.501 | 0.470 | 0.460 | 0.538 | 0.476 |
| Rational Injection Safety Index | 1.000 | 1.000 | 1.000 | 1.000 | 1.000 | 1.000 | 1.000 | 1.000 |
| Essential drugs list index | 0.970 | 0.900 | 0.954 | 0.834 | 0.950 | 0.998 | 0.974 | 0.981 |
| *IRDP* | *4.446* | *4.452* | *4.333* | *4.175* | *4.364* | *4.280* | *4.488* | *4.438* |
| *Rank* | *3* | *2* | *6* | *8* | *5* | *7* | *1* | *4* |
| **Patient-Care Indicators** | | | | | | | | |
| Consultation time index | 0.570 | 0.529 | 0.347 | 0.267 | 0.731 | 0.645 | 0.390 | 0.890 |
| Dispensing time index | 0.362 | 0.352 | 0.353 | 0.311 | 1.000 | 0.284 | 0.285 | 0.231 |
| Dispensed medicine index | 0.926 | 0.910 | 0.880 | 0.812 | 0.910 | 0.941 | 0.970 | 0.632 |
| Labelled medicine index | 0.711 | 0.212 | 0.790 | 0.918 | 0.370 | 0.720 | 0.876 | 0.933 |
| Patient's knowledge index | 0.795 | 0.670 | 0.855 | 0.790 | 0.915 | 0.840 | 0.850 | 0.555 |
| *IRPCDU* | *3.363* | *2.673* | *3.224* | *3.098* | *3.926* | *3.430* | *3.372* | *3.241* |
| *Rank* | *4* | *8* | *6* | *7* | *1* | *2* | *3* | *5* |
| **Facility Specific Indicators** | | | | | | | | |
| Index of EML | 1.000 | 1.000 | 1.000 | 1.000 | 1.000 | 1.000 | 1.000 | 1.000 |
| Index of key drugs in stock | 0.800 | 0.790 | 0.815 | 0.812 | 0.797 | 0.800 | 0.799 | 0.798 |
| *IRFSDU* | *1.800* | *1.790* | *1.815* | *1.812* | *1.797* | *1.800* | *1.799* | *1.798* |
| *Rank* | *3* | *8* | *1* | *2* | *7* | *3* | *5* | *6* |
| **Grand Total** | | | | | | | | |
| *IRDU* | *9.609* | *8.915* | *9.372* | *9.085* | *10.086* | *9.510* | *9.659* | *9.476* |
| *Rank* | *3* | *8* | *6* | *7* | *1* | *4* | *2* | *5* |

**Note:** ONRH = Orotta National Referral Hospital, HNRH = Halibet National Referral Hospital, HRRH = Hazhaz Regional Referral Hospital, GRRH = Ghindae Regional Referral Hospital, ARRH = Assab Regional Referral Hospital, KRRH = Keren Regional Referral Hospital, MRRH = Mendefera Regional Referral Hospital, BRRH = Barentu Regional Referral Hospital, EML = Essential Medicine List, IRDP = Index of Rational Drug Prescribing, IRPCDU = Index of Rational Patient- Care Drug Use, IRFSDU = Index of Rational Facility- Specific Drug Use, IRDU = Index of Rational Drug Use.

Most of the drugs (94.73%) were prescribed from the Eritrean National List of Medicines (ENLM). This was much similar with a study conducted in Eritrea [5] which was 98.83%. However, it was much higher than 42.3% reported by Nepal [21], 53% by India [26] and 81.5% by Pakistan [11]. Moreover, this finding was lower that studies conducted in North-West Ethiopia (100%) [29] and North-East Ethiopia (100%) [30]. Such high adherence and availability could be due to the centralized medicine procurement system and the regulation that prohibits the procurement of drugs outside the ENLM.

## Patient care indicators

The average consulting time in this study was 5.46 minutes (standard: greater than 10 minutes). The average consultation time varied from 2.67 minutes in Ghindae to 8.90 minutes in Barentu. This result was similar to a study conducted in Sri Lanka (5.4) [15], but lower than studies conducted in KSA (7.3) [18] and Egypt (7.1) [4]. Moreover, it was higher than 4.61 reported from Eastern Ethiopia [9], 4.1 reported from Kenya [16] and 4.7 reported from North-East Ethiopia [30]. In general, longer consultation time is necessary to assess the patient, give appropriate heath education and increases the level of physician-patient interaction

**Table 8. Country performance indicators.**

| Performance Indicators | | Country Score |
|---|---|---|
| **Drug Prescribing Indicators** | | |
| | Non polypharmacy index | 1.000 |
| | Generic name index | 0.949 |
| | Rational Antibiotic index | 0.492 |
| | Rational Injection Safety Index | 1.000 |
| | Essential drugs list index | 0.947 |
| | *IRDP* | 4.388 |
| **Patient-Care Indicators** | | |
| | Consultation time index | 0.546 |
| | Dispensing time index | 0.405 |
| | Dispensed medicine index | 0.873 |
| | Labelled medicine index | 0.682 |
| | Patient's knowledge index | 0.789 |
| | *IRPCDU* | 3.296 |
| **Facility Specific Indicators** | | |
| | Index of EML | 1.000 |
| | Index of key medicine in stock | 0.801 |
| | *IRFSDU* | 1.801 |
| **Grand Total** | | |
| | *IRDU* | 9.484 |

**Note:** EML = Essential Medicine List, IRDP = Index of Rational Drug Prescribing, IRPCDU = Index of Rational Patient- Care Drug Use, IRFSDU = Index of Rational Facility- Specific Drug Use, IRDU = Index of Rational Drug Use.

thereby improves patient satisfaction towards the healthcare system [8]. The reason behind shorter consultation time compared to the WHO optimal could be increased workload of health staff and/or not understanding communication as an important aspect of their work role.

Average dispensing time reported in this study was 36.49 seconds which was much lower compared to what the WHO recommends (greater than 90 seconds). Assab regional referral hospital is the only hospital to reach the recommended time (95.94 seconds). This lower figure was similar to the dispensing time reported in Sri Lanka (40.2 seconds) [15], Egypt (47.4 seconds) [4] and Kuwait (54.6 seconds) [17]. However it was greater than those reported in Nigeria (12.5 seconds) [31] and Brazil (17 seconds) [32]. Moreover, results from, southern Ethiopia (96.1 seconds) [12], KSA (100 seconds) [18], Zimbabwe (150 seconds) [33], Nigeria (201 seconds) [31], and India (340 seconds) [28] showed better results probably because of adequate patient to health worker ratio and good dispensing setting. Prolonged dispensing time is necessary in improving patient care as short dispensing time (less than 90 seconds) is not enough to explain every information to the patient and it's clear that patient adherence and compliance is directly proportional to dispensing time [8]. In this current study, the observed short dispensing time could be due to various reasons. First, the layout of most of the dispensaries does not allow for private pharmacist-patient interactions. Second, dispensers do not have sufficient time to explain medications to patient as they have too much workload. Furthermore, patients do not understand about the dispenser's role and they also do not expect to learn more from dispensers about drugs.

The percentage of drugs actually dispensed was 87%. This indicator was lower than the ideal WHO standard (100%). This value was similar from study done in southern Ethiopia (86.3%) [12]. However, it was lower than studies reported in Egypt (95.9%) [4], Kuwait (97.6%) [17], and KSA (99.6%) [18]. The result might be due to inadequate medicine supply and leads to unnecessary medication charge by the patients from private drug retail outlets.

An adequate label includes at least the name of the medicine, name and strength of the medicine and written instructions on how to take the medicine. The median percent medicines adequately labelled was found to be 68.24%. Lower results were observed in North-East Ethiopia (0%) [30], KSA (10%) [18], Eastern Ethiopia (20%) [9], and Kuwait (66.9%) [17]. However, Egypt (95.9%) [4] and India (100%) [26] are almost consistent with the WHO optimal value. Labelling is one of the key indicators of good dispensing practice, adequate labelling eventually promotes patient awareness about the regimen the patient takes and hence increases treatment adherence [8]. This difference could be attributed to dispenser's adequacy of training in how drugs are to be packaged and labelled. Likewise, the workload of the dispensers could also explain the inadequate labelling.

Patient's knowledge of correct dosage in this study was found to be 78.88% (optimal value: 100%). This current study reported higher patient knowledge than Kuwait (26.9) [17], India (46%) [28], Eastern Ethiopia (61.88%) [9] and KSA (79.3%) [18] but lower than the study conducted in Egypt (94%) [4]. Even though the interview was done immediately after dispensing and might not be concluded that this knowledge persists throughout the course of therapy, patient knowledge definitely improves patient care and prevents any harm related to the medication and avoids medicine overuse and abuse.

## Health facility indicators

The results for the facility indicators showed that all health facilities included (100%) had a copy of Eritrean National List of Medicine (ENLM). This value was higher than the study conducted at KSA (90%) [18], Egypt (80%) [4] and Kenya (20%) [16]. WHO recommends adherence of physicians to the medicines listed in the EML/formulary when prescribing medications in order to ensure effective health care for all [8].

Majority (80.1%) of the selected key essential medicines were in stock during the study period. This figure was higher than studies conducted at KSA (59.2%) [18], Eastern Ethiopia (66.7%) [9], and almost similar with studies conducted in Egypt (78.3%) [4] and Nigeria (83.3%) [34]. This variation could be due to the difference in study sites and inventory management. A shortage of supplies of essential medicines that treat common health problems is harmful to health status of patients, in that doctors may not be able to prescribe the correct essential medicine or they are limited to prescribing out-of-stock medicines which may pose extra financial burden on the patients' "through out of pocket" expense [8]. This requires an immediate attention from the policy makers in that essential medicines should be fully accessible without any inconsistency in their supply.

## Indices of performance indicators

This study revealed that the overall IRDP was 4.388. It was lower than the optimal value of five. Moreover, it was also lower than 4.43 reported from North-East Ethiopia [30]. However, it was higher than a study conducted in Sri Lanka (3.67) [15]. The overall IRPCDU and IRFSDU were 3.296 (out of 5.0) and 1.801 (out of 2.0), respectively. The overall IRPCDU and IRFSDU were much higher than a study conducted in North-East Ethiopia (2.51 and 0.64, respectively) [30]. This finding necessitates the policy makers and program managers to find a quick solution to improve the rational use of medicines in all areas.

## Limitations of the study

This indicator highlights general prescribing and dispensing problems at each facility. These results indicate where the problem exists but could not reveal reasons that lead to irrational use of medicines. These indicators do not show whether the prescribed medicines in the study is in compliance with the diagnosis. Besides, our findings could not be generalized to the whole of Eritrea. The authors therefore recommended further studies that explore the reasons that lead to irrational use of medicines.

## Conclusion

The overall rationality of medicine use was found sub-optimal as some of important components were missed. Three of the five prescribing indicators (percentage of encounter with antibiotic, percentage of medicines prescribed by generic and percentage of medicines from EML) and all the patient care indicators were less than the optimal value. Average number of medicines per prescription and percentage of encounters with injection prescribed falls within the window of WHO criteria. Moreover, inappropriate use of antibiotics was highly noticeable. The overall results of medicines use studies showed that there is highly appreciable practice with generic name prescribing and a high adherence to the Eritrean National List of Medicines.

## Recommendations

Considering such irrational medicine use, further in-depth investigation is warranted to dig out the underlying problem hence interventional strategies can be designed to redirect the current drug use pattern. Besides, proper utilization of standard treatment guidelines, essential medicine lists by healthcare professionals, establishment of medicines and therapeutic committee and targeted educational programs are highly recommended to enhance the rational use of medicines.

## Supporting information

**S1 File. Prescribing indicator recording form.**
(PDF)

**S2 File. Patient-care indicator recording form.**
(PDF)

**S3 File. Facility indicator recording form.**
(PDF)

## Acknowledgments

We would like to forward our sincere thanks to all medical directors of the respective hospitals for their permission and assistance during the period of this study. We also sincerely thank the data collectors, data entry personals and data editors, who contributed a lot in gathering, entry and editing of all the data required for the study. Finally, we would like to thank all the partakers of the study for being supportive in the study.

## Author Contributions

**Conceptualization:** Senai Mihreteab Siele.

**Data curation:** Senai Mihreteab Siele, Eyasu H. Tesfamariam.

**Formal analysis:** Senai Mihreteab Siele, Nuru Abdu, Eyasu H. Tesfamariam.

**Methodology:** Senai Mihreteab Siele, Nuru Abdu, Mismay Ghebrehiwet, M. Raouf Hamed, Eyasu H. Tesfamariam.

**Supervision:** Mismay Ghebrehiwet, M. Raouf Hamed, Eyasu H. Tesfamariam.

**Writing – original draft:** Senai Mihreteab Siele, Nuru Abdu.

**Writing – review & editing:** Senai Mihreteab Siele, Nuru Abdu, Mismay Ghebrehiwet, M. Raouf Hamed, Eyasu H. Tesfamariam.

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
