## [Decision Letter · Decision Letter 0]

9 Nov 2021

PONE-D-21-11930

Drug Prescribing and Dispensing Practices in Regional and National Referral Hospitals of Eritrea: Evaluation with WHO/INRUD Core Drug Use Indicators

PLOS ONE

Dear Dr. Abdu,

Thank you for submitting your manuscript to PLOS ONE. After careful consideration, we feel that it has merit but does not fully meet PLOS ONE’s publication criteria as it currently stands. In particular, the implications of the findings of the study need to be further explored and included in the discussion. Therefore, we invite you to submit a revised version of the manuscript that addresses the points raised during the review process.

Few comments related to study methods/ discussion are 

1. How was non-polypharmacy defined in the study? it is mentioned that two or less drugs were defined as non-polypharmacy. Usually polypharmacy is defined as five or more drugs in a prescription. "I*ndex of non-polypharmacy was measured by dividing average number of drugs by 1.8."* Why 1.8? and On page 7, line 136 the formula mentions observed value in the denominator.

2. What is the source of WHO standard as mentioned in the tables? The term optimal value is also used at several places in the text. What is the difference between - WHO standard and optimal values? It is worthwhile to give a table mentioning the optimal values used in the study and their references.

3. The naming of healthcare facilities needs to be uniform in the manuscript text especially tables.

4. The methods section doesn't mention about the country performance indicators. How was the value of 1 calculated for indices of non-polypharmacy and rational injection use?

5. The implications of the study findings need to be discussed in greater details for all stakeholders especially policy makers. 

6. There are some language errors and some discrepancies in the numbers mentioned in the text and table.

We look forward to receiving your revised manuscript.

Kind regards,

Ashish Kakkar, MD DM

Academic Editor

PLOS ONE

Journal Requirements:

2. Please update your Methods section to clarify if the participants gave verbal consent for the information to be used in the study --  did the trained data collectors specifically ask the patients if they consent? Also, please explain why written consent was not used.

5. We noticed you have some minor occurrence of overlapping text with the following previous publication(s), which needs to be addressed:

- http://pharmamedix.in/journals/IJPPR/article/download/107/pdf_28

- https://www.sciencedirect.com/science/article/pii/S1658361213000498?via%3Dihub

- https://bmchealthservres.biomedcentral.com/articles/10.1186/s12913-017-2097-3

In your revision ensure you cite all your sources (including your own works), and quote or rephrase any duplicated text outside the methods section. Further consideration is dependent on these concerns being addressed.

Reviewers' comments:

Reviewer's Responses to Questions

**Comments to the Author**

1. Is the manuscript technically sound, and do the data support the conclusions?

Reviewer #1: Yes

Reviewer #2: Yes

2. Has the statistical analysis been performed appropriately and rigorously? 

Reviewer #1: Yes

Reviewer #2: Yes

3. Have the authors made all data underlying the findings in their manuscript fully available?

Reviewer #1: Yes

Reviewer #2: Yes

4. Is the manuscript presented in an intelligible fashion and written in standard English?

Reviewer #1: Yes

Reviewer #2: Yes

5. Review Comments to the Author

Reviewer #1: Good day

Statistical Analysis: Kindly develop adjusted result

Ethical Approval sound appropriate. Please detail ethical approval body with address. Add Reference number of approval letter.

Please separate Conclusion and Recommendation.

Your key message in a BOX.

Please try to use recent reference add or replace more than 5 years old.

Reviewer #2: Overall

1) Correction of grammar, spellings and punctuations to improve readability.

2) I notice that the term drug and medicine are used inter-changeably, suggest sticking to the term medicine. Of course you will have to use Drug when referring to INRUD. But in other place make it uniform.

Abstract

1) Were the prescriptions collected from in-patients or out-patients or both? its not clear in the abstract

2) Which disciplines were included? coz prescribing patterns will be different. Was it all? Medicine, Surgery, Gyn & Obs, Paediatrics, etc. Again not clear in abstract.

3) In abstract SD is not reported for some parameters

4) Since most of these are likely to be skewed by outlier values, its best to report median and range as well.

Introduction

1) Would benefit from a description regarding current practices in the country and a small introduction about its health care system (public vs. private, etc, etc)

Methods

1) How big are these hospitals. An indicator about bed strength or an appropriate reference will be helpful for the reader.

2) I now see that this is mainly OPD prescriptions. However, is it all specialties?

3) How many data collectors were involved?

Results

1) As mentioned above mean may not be the best way to report. At least mention median and range within brackets

Discussion

1) The implications of the study findings are not sufficiently discussed with appropriate references. For example, use of antibiotics

2) Whats the impact of having a mixed group of patients from different specialities

3) Limitations of the study not described adequately.

6. PLOS authors have the option to publish the peer review history of their article (what does this mean?). If published, this will include your full peer review and any attached files.

Reviewer #1: **Yes: **Mainul Haque

Reviewer #2: No

---

## [Author Response · Author response to Decision Letter 0]

1 Dec 2021

Author response to Reviewer’s comments 

We would like to thank our academic editor and reviewers for their invaluable inputs and constructive comments that are helpful to massively improve the quality of the manuscript. After careful consideration of the points raised by academic editor and reviewers, a point-by-point response are as follows: 

Academic editor 

1. How was non-polypharmacy defined in the study? It is mentioned that two or less were defined as non-polypharmacy. Usually polypharmacy is defined as five or more drugs in a prescription. “Index of non-polypharmacy was measured by dividing average number of drugs by 1.8”. Why 1.8? And on page 7, line 136 the formula mentions observed value in the denominator. 

Response: It is well noted. The definition of polypharmacy in the manuscript was inappropriate and therefore omitted from the text. Besides, the statement “Index of non-polypharmacy was measured by dividing average number of drugs by 1.8” was a typing error and therefore removed from the text. Furthermore, in the result section of the study the concept of “Index of non-polypharmacy was measured by dividing the optimal value (taken as 1.8) by the average number of drugs”. The index of non-polypharmacy was expressed in terms of the average number of drugs per prescription. 

2. What is the source of WHO standard as mentioned in the tables? The term optimal value is also used at several places in the text. What is the difference between- WHO standard and optimal values? It is worthwhile to give a table mentioning the optimal values in the study and their references. 

Response: The reference for the optimal values was cited. There is no difference between the WHO standard and optimal values (used interchangeably). Moreover, an additional table (1) that mentions the WHO standard (optimal values) for all the three indicators alongside their references is added in the method section. 

3. The naming of healthcare facilities needs to be uniform in the manuscript especially tables. 

Response: Accepted 

4. The methods section doesn’t mention about the country performance indicators. How was the value of 1 calculated for indices of non-polypharmacy and rational injection use?

Response: The country performance indicators for indices of non-polypharmacy and rational injection use was calculated by the concept (optimal value/observed value). The optimal value for non-polypharmacy was taken as 1.8, whereas the optimal value for rational injection use was taken as 24.1. Moreover, the observed value was taken as the overall (average) of the hospitals in each item within the indicators. Optimal index that yielded more than one was taken as one as the maximum value of the optimal index is one. The above-mentioned calculation approach for the country performance indicators was included in the ‘method’ section within the revised manuscript. 

5. The implications of the study findings need to be discussed in greater details for all stakeholders especially policy makers. 

Response: Accepted. The implications of the study findings are discussed in greater details for all stakeholders. The discussion section is modified in the revised manuscript. 

6. There are some language errors and some discrepancies in the numbers mentioned in the text and tables. 

Response: Accepted. A massive editing is made on the manuscript. We have made 320 insertions, 236 deletions, 2 moves and 13 formatting in the revised section. Accordingly, the reference section is fully revised. Please refer the ‘revised manuscript with track changes’. 

Reviewer 1

-Statistical analysis: kindly develop adjusted result

Response: Adjusted result could not be obtained as there is only one independent variable namely, hospital. All the other variable such as average number of medicines per prescription, average consultation and dispensing time and so on are assessed on their potential difference across the hospitals. 

-Ethical approval sound appropriate. Please detail ethical approval body with address. Add Reference number of approval letter. 

Response: Accepted 

-Please separate conclusion and recommendation. 

Response: Accepted. 

-Your key message in a box. 

Response: The key message is already included in the separated recommendation section. 

-Please try to use recent references add or replace more than 5 years old. 

Response: This is a very important comment. Our paper has been prepared before 2 years. Hence, your comment is valuable and we included recent available studies in the introduction and discussion sections, while removing too old references. Besides, for WHO and other guidelines that were prepared a long time ago and which were used as a source material for the methodology we preferred to keep them in the references. 

Reviewer 2 

1. Correction of grammar, spellings and punctuations to improve readability. 

Response: Accepted. A massive editing is made on the manuscript. We have made 320 insertions, 236 deletions, 2 moves and 13 formatting in the revised section. Accordingly, the reference section is fully revised. Please refer the ‘revised manuscript with track changes’.

2. I notice that the term drug and medicine are used interchangeably, suggest sticking to the term medicine. Of course you will have to use Drug when referring to INRUD. But in other place make it uniform. 

Response: Accepted. Where appropriate, the term ‘drug’ is replaced with the term ‘medicine’. 

Abstract 

1. Were the prescriptions collected from in-patients or out-patients or both? It’s not clear in the abstract. 

Response: It was from out-patients. 

2. Which disciplines were included? Coz prescribing patterns will be different. Was it all? Medicine, Surgery, Gyn & Obs, Paediatrics, etc. Again not clear in the abstract. 

Response: All medical specialties (disciplines) were included. 

3. In abstract SD is not reported for some parameters. 

Response: Accepted. Standard deviation is reported for all parameters expressed as M (SD). 

4. Since most of these are likely to be skewed by outlier values, its best to report median and range as well. 

Response: Accepted. We reported it as mean for ease of comparison of the values with that of the WHO standard. Besides, at times median value is important we reported it. For instance, average consultation and dispensing time. 

Introduction 

1. Would benefit from a description regarding current practices in the country and a small introduction about its healthcare system (public vs. private, etc, etc)

Response: Accepted. The current practices in the country is included in the introduction. Moreover, a small introduction about the Eritrean healthcare system is given in the method section (study design and setting sub-section). 

Methods

1. How big are these hospitals? An indicator about bed strength or an appropriate reference will be helpful for the reader. 

Response: All the study sites (hospitals) are one of the biggest hospitals in the country. Orotta and Halibet hospitals are national referral hospitals that serves patients from all around the country. They are situated in the capital city of Eritrea (Asmara) and considered as a tertiary level services that provides lots of services such as out-patient, in-patient, maternity, paediatrics, and surgery and so on. Furthermore, the rest six hospitals are regional referral hospitals that serves patients referred or self-referred within the region they locate. They are considered as secondary level of service and are bigger in nature. 

2. I now see that this is mainly OPD prescriptions. However, is it all specialties?

Response: Yes, it was from all specialties. 

3. How many data collectors were involved? 

Response: A total of 20 data collectors were involved in the study. 

Results 

1. As mentioned above mean may not be the best way to report. At least mention median and range within brackets. 

Response: Accepted. At times Md (IQR) is important we reported it in some parameters like average number of medicines per prescription, average consultation and dispensing time. 

Discussion 

1. The implications of the study findings are not sufficiently discussed with appropriate references. For example, use of antibiotics. 

Response: Accepted. The implications of the study findings are expressed in detail in the revised manuscript. 

2. What is the impact of having a mixed group of patients from different specialties? 

Response: It helps to get a complete picture of the rational use of medicines within a hospital.

3. Limitations of the study not described adequately. 

Response: Accepted. We explored the possible limitations we have in the study and added in the revised manuscript. 

Kind regards 

Nuru Abdu (BPharm)

On behalf of the authors

---

## [Decision Letter · Decision Letter 1]

22 Jun 2022

PONE-D-21-11930R1Drug Prescribing and Dispensing Practices in Regional and National Referral Hospitals of Eritrea: Evaluation with WHO/INRUD Core Drug Use IndicatorsPLOS ONE

Dear Dr. Abdu,

Thank you for submitting your manuscript to PLOS ONE. After careful consideration, we feel that it has merit but does not fully meet PLOS ONE’s publication criteria as it currently stands. Therefore, we invite you to submit a revised version of the manuscript that addresses the points raised during the review process.

We look forward to receiving your revised manuscript.

Kind regards,

Ashish Kakkar, MD DM

Academic Editor

PLOS ONE

Journal Requirements:

Additional Editor Comments (if provided):

The authors have addressed majority of the reviewers' and editor's comments.

Few minor comments:

1. The tables should include SD wherever mean is mentioned. Either of them (Mean SD/ Median IQR) be reported for each variable depending upon distribution of data values.

2. Table 1 has typo. Percent of drugs adequately labelled!

3. There is discrepancy in the average number of drugs per encounter mentioned in abstract, main results, discussion and the table 2. Needs to be rechecked

4. Several statements in the discussion lack appropriate references viz. irrational and overuse of antibiotics...... , injectable preparations....

5. Limitations need to include implications of having a mixed group of patients from various specialties.

6. Discussion needs to be rechecked for consistency and appropriate references.

Reviewers' comments:

Reviewer's Responses to Questions

**Comments to the Author**

1. If the authors have adequately addressed your comments raised in a previous round of review and you feel that this manuscript is now acceptable for publication, you may indicate that here to bypass the “Comments to the Author” section, enter your conflict of interest statement in the “Confidential to Editor” section, and submit your "Accept" recommendation.

Reviewer #1: All comments have been addressed

Reviewer #3: All comments have been addressed

2. Is the manuscript technically sound, and do the data support the conclusions?

Reviewer #1: Yes

Reviewer #3: Yes

3. Has the statistical analysis been performed appropriately and rigorously? 

Reviewer #1: Yes

Reviewer #3: Yes

4. Have the authors made all data underlying the findings in their manuscript fully available?

Reviewer #1: Yes

Reviewer #3: Yes

5. Is the manuscript presented in an intelligible fashion and written in standard English?

Reviewer #1: Yes

Reviewer #3: No

6. Review Comments to the Author

Reviewer #1: Accept. As Comments are addressed I would to recommend the should be accepted and published in Plos One

Reviewer #3: The study addresses an important health care issue of the developing nations which can be controlled with appropriate measures. Prescription, patient care and facility indicators are good source of information about the rational use of medicines in a particular region which is the primary objective of the study. Keeping WHO indices as optimum values for comparison of the rational medicine use parameters is appreciable, though these values are described differently in several published papers. Though several similar studies have been reported earlier, region specific knowledge would be useful to plan further interventions in improving the deficient indices in Eritrea. The calculation of indices for the indicators and ranking the facilities based on the indices is a good initiative which only a few papers have reported earlier. The authors have answered and followed the editor and reviewer comments to a large extent. However a few more corrections are required in the language and grammar to make it easy for the readers to understand in the first read. Some of these corrections are listed below:

Introduction: Line 72 must be 'ensuring'

Methods: Line 80- ' September 1 to January 31, 2018' year for September to be mentioned

Tables 2, 5 : The abbreviations M, Md, IQR to be expanded in foot note. Column headings could be made appropriate for all the values described in the table. Values in Table 4 represent time, but the column heading does not mention time.

References:

Many are incomplete, example - Ref 1 is incomplete. Web pages are without a url.

7. PLOS authors have the option to publish the peer review history of their article (what does this mean?). If published, this will include your full peer review and any attached files.

Reviewer #1: **Yes: **Mainul Haque

Reviewer #3: **Yes: **Jayanthi M

---

## [Author Response · Author response to Decision Letter 1]

25 Jun 2022

Author response to editor’s and reviewer’s comments 

We would like to thank our academic editor and reviewers for their invaluable inputs and constructive comments that are helpful to massively improve the quality of the manuscript. After careful consideration of the points raised by academic editor and reviewers, a point-by-point response are as follows: 

Academic editor 

1. The tables should include SD wherever mean is mentioned. Either of them (Mean SD/Median IQR) be reported for each variable depending upon distribution of data values. 

Response: Thank you for this comment. SD is included wherever mean is mentioned. Besides, the authors preferred to report both the mean (SD) and median (IQR). Mean (SD) is used for ease of comparison with the WHO standard values and median (IQR) is included as some variables have outliers. 

2. Table 1 has typo. Percent of drugs adequately labelled!

Response: Thank you for pointing out this issue. The typos error in table 1 is corrected in the revised manuscript. 

3. There is discrepancy in the average number of drugs per encounter mentioned in the abstract, main results, discussion and the table 2. Needs to be rechecked. 

Response: Comment is addressed in the revised manuscript. 

4. Several statements in the discussion lack appropriate references viz. irrational and overuse of antibiotics……., injectable preparations……….. 

Response: Thank you for the great comment. The discussion section is revised and where necessary citations are added. Besides, some statements used as implications are the authors’ own ideas. 

5. Limitations need to include implications of having a mixed group of patients from various specialties. 

Response: Thank you for this comment. The study sites are regional and national referral hospitals having various specialties (disciplines). Thus, the study involved a mixed group of patients. So, the authors do not see the importance of adding it in the limitation section. 

6. Discussion needs to be rechecked for consistency and appropriate references. 

Response: Comment well noted. Necessary changes are made in the revised manuscript. 

Reviewer 1

-Accept. As comments are addressed, I would like to recommend they should be accepted and published in PLOS ONE. 

Response: Thank you for your review. 

Reviewer 2 

-Introduction: Line 72 must be ‘ensuring’

Response: Thank you for your comment. It was corrected in the revised manuscript. 

-Methods: Line 80- ‘September 1 to January 31, 2018’ year for September to be mentioned. 

Response: Comment well taken. Addressed in the revised manuscript. 

-Tables 2, 5: The abbreviations M, Md, IQR to be expanded in foot note. 

Response: Comment well taken. Addressed in the revised manuscript. 

-Column headings could be made appropriate for all the values described in table. Values in Table 4 represent time, but the column heading does not mention time. 

Response: Thank you for your suggestion. For ease of understanding, the authors added ‘minute’ and ‘second’ alongside the values for the consultation and dispensing time, respectively. 

-Many are incomplete, example – Ref 1 is incomplete. Web pages are without a URL. 

Response: Thank you for pointing out this issue. References are updated in the revised version. 

Kind regards 

Nuru Abdu (BPharm)

On behalf of the authors

---

## [Editor Report · Decision Letter 2]

1 Aug 2022

Drug Prescribing and Dispensing Practices in Regional and National Referral Hospitals of Eritrea: Evaluation with WHO/INRUD Core Drug Use Indicators

PONE-D-21-11930R2

Dear Dr. Abdu,

We’re pleased to inform you that your manuscript has been judged scientifically suitable for publication and will be formally accepted for publication once it meets all outstanding technical requirements.

Kind regards,

Ashish Kakkar, MD DM

Academic Editor

PLOS ONE

Additional Editor Comments (optional):

The authors need to correct a minor typo in Table 1 that was mentioned in the comments - "Percent of drugs "actually" labelled. - It should be "adequately" labelled.
---

## [Editor Report · Acceptance letter]

10 Aug 2022

PONE-D-21-11930R2 

Drug Prescribing and Dispensing Practices in Regional and National Referral Hospitals of Eritrea: Evaluation with WHO/INRUD Core Drug Use Indicators 

Dear Dr. Abdu:

I'm pleased to inform you that your manuscript has been deemed suitable for publication in PLOS ONE. Congratulations! Your manuscript is now with our production department. 

Kind regards, 

on behalf of

Dr. Ashish Kakkar 

Academic Editor

PLOS ONE